# Nurse-Led Interventions for Improving Medication Adherence in Chronic Diseases: A Systematic Review

**DOI:** 10.3390/healthcare12232337

**Published:** 2024-11-22

**Authors:** Daniela Berardinelli, Alessio Conti, Anis Hasnaoui, Elena Casabona, Barbara Martin, Sara Campagna, Valerio Dimonte

**Affiliations:** 1Department of Public Health and Pediatrics, University of Torino, 10126 Torino, Italy; daniela.berardinelli@unito.it (D.B.); elena.casabona@unito.it (E.C.); valerio.dimonte@unito.it (V.D.); 2Department of Clinical and Biological Sciences, University of Torino, 10126 Torino, Italy; alessio.conti@unito.it; 3Faculty of Medicine of Tunis, Tunis El Manar University, Rue Djebal Lakhdar, Tunis 1006, Tunisia; anis.hasnaoui@fmt.utm.tn; 4Signals and Smart Systems Lab L3S, National Engineering School of Tunis, Tunis El Manar University, Campus Universitaire Farhat Hached B.P. n° 94-ROMMANA, Tunis 1068, Tunisia; 5General Affairs and Cultural Heritage Directorate, University of Torino, 10126 Torino, Italy; barbara.martin@unito.it

**Keywords:** chronic disease, medication adherence, medication review, nurse-patient relations, patient compliance, self medication

## Abstract

**Background**: Poor medication adherence results in negative health outcomes and increased healthcare costs. Several healthcare professionals provide interventions to improve medication adherence, with the effectiveness of nurse-led interventions in people with chronic diseases remaining unclear. **Objective**: This study sought to evaluate the effectiveness of nurse-led interventions for improving medication adherence in adults with chronic conditions. **Methods**: Five databases (MEDLINE, CINAHL, EMBASE, Cochrane Library, SCOPUS) were searched without applying a temporal limit. Studies evaluating the effects of nurse-led interventions on medication adherence in adults with one or multiple chronic conditions were included. Interventions only targeting a single acute disease were excluded. **Results**: A total of twenty-two studies with 5975 participants were included. Statistically significant improvements in adherence were reported in five out of seven studies involving face-to-face visits to patients with heart failure (n = 2), chronic myeloid leukemia (n = 1), hypertension (n = 1) and multimorbidity (n = 1) and in four out of nine studies adopting a mixed method involving face-to-face visits and telephone follow-up for patients with heart failure (n = 1), hypertension (n = 1), coronary disease (n = 1) and multimorbidity (n = 1). Remote interventions were effective in improving medication adherence in one out of six studies. No statistically significant differences were found between tablet computer-based patient education and nurse-led educational sessions. The motivational approach was found to be one of the most common strategies used to promote patient medication adherence. **Conclusions**: Nurse-led face-to-face visits may be effective in improving medication adherence in people with chronic diseases. However, further research is needed because current methods for measuring medication adherence may not accurately capture patient behaviour and medication consumption patterns.

## 1. Introduction

The World Health Organization (WHO) defined medication adherence as “the extent to which a person takes medication, corresponds to agreed recommendations from a health care provider” [1]. Medication adherence is crucial for the success of treatment plans and improving patient outcomes, yet adherence to long-term therapy for chronic illnesses is estimated at only around 50% and is a common problem that occurs in various contexts, regardless of the disease being treated, its severity, or access to healthcare resources [1,2].

Many patients do not completely comply with their prescribed medications and often discontinue or do not take them as prescribed after the first month [3]. Poor medication adherence therefore represents a major obstacle to realizing the benefits of drugs that have proven to be more beneficial than harmful in clinical trials [3].

Improving medication adherence can reduce morbidity, mortality, and inpatient hospital stays and improve overall health condition [4,5,6]. Conversely, poor medication adherence results in negative health outcomes and increased healthcare costs [4]. Numerous factors are known to be associated with poor adherence, including patient demographics, psychosocial and socioeconomic factors, disease-related issues, and the patient–provider relationship [7,8].

Psychosocial and cultural beliefs can influence a patient’s medication behaviour, for example, a sense of mistrust or negative beliefs about medication, which may lead to thinking that the medication is ineffective or not really needed when symptoms resolve or when a potential side effect is experienced [8]. Such beliefs could be changed through education and support provided by healthcare professionals [8].

Non-adherence to medication has been defined as both “taking less than 80% of the prescribed doses” or equally taking higher doses [3]. Risk factors for non-adherence include polypharmacy, complex treatment plans, cognitive decline, and lack of support from health professionals and family members [6,8]. Polypharmacy is particularly problematic in older adults who often have multiple chronic conditions [6,9].

Although no optimal strategies for enhancing medication adherence have been recognized, building a strong alliance, educating patients, involving caregivers, and monitoring tolerability and side effects could improve patients’ medication adherence [10]. Consequently, all the patient’s disease-related and socioeconomic factors should be explored within the context of their lived experience. Notable too is that patients demonstrate better adherence when receiving care from the same provider over time, with attention paid to tailoring the communication of their needs [1].

Intervention to improve medication adherence can be provided by various healthcare professionals such as physicians, nurses, and pharmacists, as highlighted in previous systematic reviews [5,6,7,11]. A previous Cochrane systematic review [3] examined a variety of interventions provided by different health professionals aimed at improving medication adherence and concluded that there is insufficient evidence to suggest that medication adherence can be effectively enhanced with the current resources available in clinical settings. A proposed solution to the problem involves expanding the nursing role by allocating specific time for medication education and management to promote adherence in a sustainable and feasible manner within clinical settings [3]. Even if nurses are the most numerous healthcare professionals and closest to the patient in terms of knowledge and daily interaction, there is limited evidence regarding the impact of nursing interventions, either alone or in conjunction with other healthcare professionals, on improving medication adherence among adults with chronic conditions.

As frontline healthcare providers, nurses are uniquely positioned to understand patients’ needs and challenges daily, making them valuable contributors to policymaking and decision-making processes [12,13]. By actively participating in developing and implementing health policies, nurses can advocate for improvements in patient care, access to healthcare services, and overall health outcomes [14]. Exploring the impact of nursing intervention is fundamental as they already deal with medication administration and education in the healthcare environment daily [15].

The only available evidence supporting the effectiveness of nurse-led intervention in improving medication adherence is inconsistent, focused on specific diseases and generally in combination with psychological interventions [5,7,11]. In fact, two systematic reviews reported that nurse-led motivational interviewing and counselling enhanced medication adherence only in individuals with heart disease, older adults discharged from hospitals, and those with HIV [5,7,11].

To our knowledge, no previous systematic reviews have analyzed the effectiveness of nurse-led interventions alone in improving medication adherence in people with one or multiple chronic diseases.

Considering the impact of poor or non-adherence to medications on health outcomes in people with chronic conditions and on health services, this systematic review aimed to synthesize evidence on the effectiveness of nurse-led interventions in improving medication adherence in adults with chronic conditions. In addition, evidence-based strategies to enhance medication adherence and health outcomes in people with chronic diseases have been highlighted.

## 2. Methods

### 2.1. Design

We conducted a systematic review and a meta-analysis according to the updated Preferred Reporting Items for Systematic Reviews and Meta-Analyses (PRISMA) guidelines [16]. The review protocol was registered in the PROSPERO register of systematic reviews on 10 March 2023 (registration number CRD42023403467), available at https://www.crd.york.ac.uk/prospero/display_record.php?RecordID=403467 (accessed on 10 October 2024). We made no amendments to the original protocol.

### 2.2. Search Strategy

To create a comprehensive search strategy and identify keywords, we searched five databases (MEDLINE via Pubmed, CINAHL, EMBASE, Cochrane Library, and SCOPUS) initially from their inception to 1 March 2023, with the search re-run on 17 May 2024. One investigator (DB) with experience searching the literature under a health librarian’s (BM) supervision conducted this search. A combination of free text and MeSH terms were used, and no temporal restrictions were applied. An example search strategy is shown in Table 1. The full search strategy is presented as a Appendix A. In addition, the PROSPERO register of systematic reviews was searched for ongoing, recently completed reviews and clinical trial registers (WHO International Clinical Trial Register Platform and Clinicaltrial.gov).

### 2.3. Eligibility Criteria

We included clinical trials, controlled clinical trials, and randomized controlled trials published in peer-reviewed journals that assessed the efficacy of interventions performed by nurses on patients with one or more chronic condition. To be eligible, studies had to provide nurse-led or nurse-collaborative interventions to adults (≥18 years) and measure medication adherence as a primary or secondary outcome. Interventions were defined as such if they were based on a theory, a theoretical framework, a model, or an explicit method. We included nurse-collaborative interventions only where data related to the nurse’s role were described. Interventions were defined as nurse-collaborative where their involvement was part of an interdisciplinary team. Nurse-led or nurse-collaborative interventions were compared to usual care or any other control interventions. Interventions only targeting an acute single disease were excluded. Studies that focused on addiction diseases, where adherence problems are typically of a different nature and much more severe, as well as studies reporting patients’ lived experience during nurse-led interventions and follow-up were excluded. In addition, conference proceedings, theses, letters to the editor, and other grey literature were excluded.

### 2.4. Article Screening and Study Selection

Two reviewers (DB, AC) independently screened the titles and abstracts of all records to remove duplicates and identify relevant publications. Rayyan software (https://www.rayyan.ai/) was used for the title/abstract screening process. We retrieved the full-text versions of all potentially relevant records. In case of disagreements, a third reviewer (SC) was involved.

### 2.5. Assessment of Risk of Bias

The risk of bias was independently assessed by two reviewers (DB, AH) using the Revised Cochrane risk-of-bias tool for randomized trials (RoB2) [17]. Any disagreement was solved through discussion with a third investigator (AC). RoB 2 is the recommended tool to assess the risk of bias in randomized trials. RoB 2 is structured into a fixed set of 5 bias domains, focusing on different aspects of trial design, conduct, and reporting. Each domain provides a series of questions that aim to elicit information about features of the trial that are relevant to the risk of bias, with five response options (yes, probably yes, probably no, no, and no information). A judgement about the risk of bias arising from each domain is proposed by an algorithm based on given answers, but this needs to be confirmed by the investigator. The final judgement can be “Low” or “High” risk of bias, or it can express “Some concerns”. The overall risk of bias was rated as “low” if the risk of bias was low in all key domains. The study is judged to raise “some concerns” if it is in at least one domain for this result, but not to be at high risk of bias for any domain. The study is judged to be at high risk of bias in at least one domain for this result, or the study is judged to have some concerns for multiple domains in a way that substantially influences the result. The suggested Excel tool was used to collect risk-of-bias assessment data and generate visual “traffic light” plots of each domain-level judgement and overall judgement (available at riskofbias.info).

### 2.6. Data Extraction

Data were extracted by two independent reviewers (D.B. and A.C.). Data on study characteristics (author, country, year, study design, setting, study sample, description of the intervention, assessment tools or instruments, primary outcome investigated and timing of outcome measurements and follow-up), participants (sample size, chronic conditions), narrative summary of findings, and quantitative results (means and standard deviations or frequencies and percentages) were entered into a data collection form (Excel spreadsheet).

Discrepancies were resolved through discussion and involving a third reviewer (S.C.) when necessary. Efforts were made to contact authors to obtain missing information. The data collection form was piloted in five studies, and appropriate improvements were made.

### 2.7. Primary Outcome

Our primary outcome was patient medication adherence measured directly or indirectly. Direct methods included patient observation, taking the medication, or measuring medication levels in plasma or urine. Indirect methods encompassed pill count, use of pharmacological databases, a medication event-monitoring system (MEMS), questionnaires, scales, or self-reporting by patients or health professionals [18].

### 2.8. Secondary Outcomes

Our secondary outcomes were mortality, hospital readmission, need of urgent care and potentially relevant clinical outcomes.

### 2.9. Data Synthesis

Studies were grouped according to the mode of delivery and the characteristics of the intervention. Based on the Cochrane paradigm of Richards et al. [19], interventions were classified into two main categories of delivery: face-to-face or remote. Face-to-face interventions are characterized by the interaction between the implementer and the participant, which takes place in person. Instead, remote modalities included web-based, telephone-based, and telemonitoring interventions [20].

## 3. Results

### 3.1. Articles Included in Systematic Review and Meta-Analysis/Search Results

The search strategy generated 6137 articles, of which 1217 were duplicates which were removed. Screening of the titles and abstracts led to the exclusion of 4480 articles. A total of 40 articles were taken forward for full-text review. Eighteen were excluded as they did not meet the inclusion criteria, as stated in Figure 1. After re-running the search strategy, no additional articles were found. Twenty-two articles are included in this review. Among those included, additional data were requested from seven authors for data conversion for meta-analysis [21,22,23,24,25,26,27]. Only two authors answered, providing the requested information [22,24].

### 3.2. Characteristics of Studies Included in the Systematic Review

All studies were randomized control trials [21,22,23,24,25,26,26,27,28,29,30,31,32,33,34,35,36,37,38,39,40,41,42]. Specifically, one was a cluster [37] and one was a pragmatic randomized control trial [31]. Blinding of the participants was not possible due to the nature of the interventions. Table 2 lists the characteristics of the included studies. These studies were published from 2006 to 2022 and performed in China [27,29,30,32,33,39,40,41,42], in the USA [22,23,24,34,37], in Europe [21,25,26,31,38], and in South America [28,35,36]. In sixteen studies, medication adherence was the primary outcome [21,22,23,24,25,27,28,29,32,34,36,38,39,40,41,42]. Other included outcomes were clinical endpoints, clinical outcomes, and patient-reported outcomes. Clinical endpoints were mortality [21,30,33,40,42], hospital readmission [21,30,32,33,39,40,42], and need of urgent care [21,22,33,39,41]. Clinical outcomes included blood pressure [27,29,35,36,37], BMI [35], immunosuppression level [26], glycated hemoglobin [31,36] and cholesterol level [31]. Patient-reported outcomes were self-care [28,40], self-management [27,30,35,37,41], self-monitoring [34,39], illness perception [22], coping strategies [32], knowledge of drug indications [37,41], treatment burden [41], QOL [27,27,31,32,35,37,39,40,42], healthy lifestyle behaviours [29,31], psychosocial outcomes [31], and satisfaction with care [22,39,41].

### 3.3. Participants and Settings

The study sample size was variable, ranging from 33 [38] to 2665 [31], with a total of 5975 participants included in this review. The mean age of the individuals in the included studies was 59 years and ranged from 38 [38] to 82 [21]. Five studies targeted participants with multimorbidity, namely diabetes, depression and coronary disease [34]; hypertension and type 2 diabetes [36]; diabetes, hypertension, coronary heart disease, heart failure, depression and schizophrenia [31]; and hypertension, coronary heart disease, stroke and cerebrovascular disease [41]. One study did not state patients’ type of chronic disease [33]. The remaining studies were conducted among participants with a single chronic disease, such as heart failure [23,28,30,40,42], hypertension [27,29,35,37], coronary disease [21,22], atrial fibrillation [32], chronic myeloid leukemia [25], kidney disease [39], schizophrenia [38], HIV [24], and lung transplantation [26]. The studies were undertaken in a variety of different settings, such as hospital [21,22,23,25,26,27,29,30,33,39,40,42], health centres [32,36,37,41], clinics [24,28,34,35] and in ambulatory care [38]. For one study, data were retrieved from an insurance registry [31].

### 3.4. Characteristics of the Nurse-Led Intervention

Interventions were nurse-led in twenty studies and nurse-collaborative in two [34,37]. The detailed characteristics of the nurse-led interventions were extracted and are summarized in Table 3. In seven studies, nurse-led interventions were delivered through face-to-face visits [23,25,26,27,28,36,40]; in six studies, they were delivered through remote modalities (i.e., telephone follow-up or telemonitoring) [22,31,32,38,39,42]; and in nine studies, the two delivery methods were combined [21,24,29,30,33,34,35,37,41]. Among interventions carried out face-to-face, fourteen were structured educational interventions [21,23,24,25,27,28,29,30,34,35,36,37,40,41]. The frequency of these interventions varied from one meeting or encounter [21,25] to a maximum of eight [28], with their duration ranging from a five-minute telephone call [35] to a one-hour face-to-face visit [35]. Follow-up periods ranged from 24 h after the discharge of the patient [22] to 36 months [31] after the intervention.

Twenty studies compared nurse-led interventions with usual care [21,22,23,24,25,27,28,29,30,31,33,34,35,36,37,38,39,40,41,42], whereas two studies compared nurse-led intervention with a web-based integrated management programme using tablet- and computer-based patient education [26,32].

### 3.5. Medication Adherence Measurements

Measurements of medication adherence varied widely (Table 2). All studies used indirect detection methods of medication adherence except for one, which additionally used a direct measure [26]. Multiple indirect methods were used to measure medication adherence: self-report scales [21,22,23,24,25,26,27,28,30,31,32,33,35,36,39,40,41], pill count [21,23,24,37], pharmacy refill records [24,34,42], and MEMS [24,38]. Among the thirteen different self-report scales used in the included studies, the Morisky Medication Adherence Scale (MMAS) [21,22,23,24,25,26,27,40] and the Medication Adherence Report Scale (MARS-5) [31,41] were the most frequently used (Table 2). Three studies adopted a mix of self-report and pill count [21,23] or self-report, pill count and pharmacy refill data [24].

### 3.6. Effects on Medication Adherence

Only four studies [21,30,35,41] highlighted a statistically significant increase in medication adherence using face-to-face visits plus telephone follow-up (Table 2). However, five studies found that face-to-face visits alone were statistically significant [23,25,27,36,40]. Among the face-to-face interventions, a statistically significant improvement in medication adherence was found in three studies, including patients with heart failure [23,30,40], two with multimorbidity [36,41], two with hypertension [27,35], one with coronary disease [21], and one with chronic myeloid leukemia [25].

Of the six studies offering a remote delivery intervention [22,31,32,38,39,42], only the study by Hsieh et al. [32] found a statistically significant improvement in medication adherence. In one study, no statistically significant differences were found between tablet computer-based patient education and one nurse educational session [26].

### 3.7. Effect of Nurse-Led Interventions on Other Outcomes

The nurse-led intervention also impacted other outcomes, such as mortality (OR 0.37, 95% CI = 0.154–0.892, *p* = 0.027) [33], hospital readmission (I = 10.4% C = 27.1% *p* 0.036; I = 6.38% C = 23.91% *p* = 0.038; 8.0% vs. 5.2% per person-week; OR 0.406; 95% CI 0.178–0.926; *p* = 0.03) [30,32,40,42], urgent care (OR = 0.388, 95% CI = 0.183–0.822, *p* = 0.013) [33], and systolic blood pressure, with a reduction ranging from 3 mmHg to 19 mmHg [27,29,35,36,37].

### 3.8. The Methodological Quality of the Included Studies

The quality of the studies was suboptimal: out of 22 studies, 1 was at low risk, 15 showed some concerns, and 6 were at high risk of bias (Figure 2). Although all studies were randomized controlled trials, only nine detailed the allocation concealment process [21,26,28,29,32,35,36,37,41]. Bias due to missing outcome data was of concern in ten studies [21,22,24,28,31,33,37,38,39,41]. Sixteen studies showed some concern or a high risk of bias in the measurement of the outcome [22,23,24,25,27,29,30,31,32,33,34,35,37,40,41,42]. This was due to a lack of validated measurement methods for the outcome and a lack of blindness that may have caused a recall or a social desirability bias and a self-report bias.

## 4. Discussion

This systematic review aimed to synthesize the evidence on the effectiveness of nurse-led interventions that set out to improve medication adherence in adults with chronic conditions. The evidence of effectiveness presented in this systematic review was mixed and inconsistent, probably due to the different chronic conditions and medication adherence measurement methods, making comparing results challenging. Similarly, significant variability in data presentation was found in a previous systematic review conducted on nurse-led interventions [5]. Due to this heterogeneity, it was not possible to perform a meta-analysis.

Substantial differences in intervention conditions (such as type, frequency, duration, medication, adherence measures, and outcome composition) contributed to high heterogeneity. Other systematic reviews of adherence interventions have also noted this limitation [3,5,6,11]. Adherence is prevalently measured with indirect methods. This, in combination with the use of thirteen different questionnaires to detect medication adherence, makes comparability between studies challenging. The use of self-report questionnaires could also overestimate the impact of interventions [3]. While patient perspectives are important for enhancing patient empowerment in managing drug therapy and maintaining adherence, combining subjective data with objective measures using alternative methods of measuring medication adherence would be appropriate [43].

In our review, three studies used a combination of self-report questionnaires with objective measures such as pill counts, MEMS, and pharmacy refill data [21,24,37]. Employing a combined objective and subjective approach to measuring adherence could strengthen results, but methods like pill counts and MEMS have limitations. Pill counts require patients to return containers with unused medication and can be easily manipulated by patients overestimating their drug intake [5]. Additionally, pill counts only provide an estimate of adherence and do not offer insight into patients’ drug-taking behaviours [18]. Many studies used electronic monitoring with MEMS cups to generate objective data on the timing of medication intake by recording bottle cap openings, but this method does not confirm whether the patient actually took the medication [3]. Automated pharmacy refill data constitute another objective measure that ensures patient and provider blinding but does not provide information on patient behaviours [3]. The wide range of methods used to measure medication adherence is a well-known issue, as there is still no consensus on the best approach [3].

Our results are coherent with other systematic reviews conducted on people with multimorbidity, highlighting the potential role of nurse-led intervention in improving medication adherence [5,44]. Among all the considered interventions, slight medication adherence improvements were observed in face-to-face nurse-led visits [23,35,36,41]. The structure of nurse-led face-to-face interventions is complex and multi-component. In fact, different resources for educational sessions of the included studies were involved (i.e., verbal education, written material, technological support), increasing the knowledge and awareness of patients on the importance of medication adherence. During these educational sessions, nurses adopted motivational techniques to assist patients in changing negative behaviours and beliefs regarding medications and addressing patient doubts and concerns [36,41].

Patient motivation is a key factor in medication adherence, reflecting the patient’s willingness to modify behaviours and thoughts. Motivational interviewing involves a collaborative effort between the healthcare professional and the patient to identify goals that will be pursued and guide future sessions [45]. Implementing an individualized plan tailored to the patient’s specific needs [23,35,36,41] was frequently adopted to set specific goals with patients, focusing on lifestyle modifications and the effectiveness of treatments and their side effects. A systematic review focusing on cardiovascular patients indicates that motivational interviewing is a promising intervention for nurses to enhance medication adherence by improving patient capabilities, confidence, and motivation to achieve mutually agreed-upon goals [7]. A meta-analysis examining the effectiveness of motivational interviewing interventions on medication adherence in adults with chronic diseases found a positive effect, albeit small [46].

Furthermore, the motivational approach is one of the most common strategies used to promote patient self-management for people with chronic diseases in primary care [47]. The few nurse-led face-to-face educational interventions were designed to enhance understanding of the disease, provide self-care management measures and promote lifestyle modification strategy [23,41]. Self-management education and support (SMES) programmes are recommended to enhance the quality of life of individuals with chronic conditions and guide their health-related decisions and activities [47]. Nurses were the main healthcare professionals involved in the SMES programmes, playing a key role in promoting self-management [47].

Since medication adherence is a life-long phenomenon in people with chronic diseases, one might assume that nurse-led interventions aimed at improving medication adherence should be of a long duration; otherwise, their efficacy could be lost. In our review, medication adherence tends to be higher at the conclusion of the intervention and declines once the intervention is terminated [36,41], highlighting the benefits of maintaining constant follow-up over time [48]. However, the optimal duration, frequency and follow-up of nurse-led interventions still remain unclear. In our systematic review, a few studies had short follow-up periods, and their long-term efficacy should be further explored [32,41]. As adherence is an ongoing process for individuals with chronic conditions, this would suggest that medication adherence requires consistent monitoring and evaluation over time. However, a recent and unrelated study assessed the maintenance of effectiveness of a lifestyle counselling intervention over five years aimed at reducing blood pressure. While the intervention was effective at year one, its effectiveness waned after the educational intervention was discontinued following the final follow-up at the end of this year [49]. Evidence suggests that involving patients and making them partners in care has the potential to improve health outcomes and enhance healthy behaviours [50].

In our systematic review, an important factor found to influence medication adherence was the support of family members and caregivers in managing therapy and in recognizing and managing signs and symptoms [23,35]. In accordance with this, a meta-analysis revealed that social support from families, friends, and healthcare professionals was significantly linked to medication adherence [51]. Exploring the family context and living arrangements might therefore enhance medication adherence and identify other factors that may impact it or other healthcare needs. Socioeconomic conditions may also play a role in medication adherence, but neither our studies nor previous reviews have considered them as influencing factors [8].

Besides the support of family members, one study showed the effectiveness of nurse-led group educational interventions with patients with the same disease [35]. Currently, there is insufficient evidence to determine whether combining group and individual educational interventions is more effective [52]. Nevertheless, this creates an opportunity to consider educating more patients simultaneously and encourage relational exchange and sharing of self-management strategies.

Our results showed that most interventions improving medication adherence occurred in primary care settings [35,36,41], whereas only one intervention started at the patient’s discharge [23]. Interventions initiated at the hospital community level may be most likely to influence medication adherence [4]. Establishing a link between the hospital and primary care would be beneficial for maintaining continuity of care for patients. The primary care outpatient clinic, which deals with patients who are not hospitalized and, therefore, more clinically stable, should also provide more dedicated time to conduct a thorough care assessment of patients [53]. Involving nurses in advanced roles is a recognized way to optimize healthcare resources and increase quality of care. It is believed that the inclusion of nurses in a pivotal role can ensure that the demand for healthcare services that meet the needs of patients is adequately met [54]. Organizations such as the World Health Organization (WHO) have made several recommendations on how to strengthen the role of nurses (WHO 2012), and policy makers believe that in order to meet the challenges of hospitals and primary care, a more structured workforce in healthcare is required [55]. Nurse-led interventions lead to better health outcomes for a wide range of patient conditions, and the empowerment of the nursing role is vital and essential in the management of chronic diseases. With regard to the effects on clinical practice, despite the large heterogeneity of inventions, this study provides a wide view of nurse-led educational interventions that nurses could carry out both in community and hospital care. Practical implications for health systems and policy makers could be the integration of nurses-led clinics for the management of chronic diseases. The burden of complex and chronic diseases is increasing and requires expert nurses and financial resources to manage chronic and multimorbid patients properly. Nurse-led clinics could ensure the management of people with chronic diseases and help identify patients with unmet needs or complications early, giving appropriate care for patients and reducing pressure on the health system. Nurse-led clinics could represent a bridge between the hospital and the community and ensure continuity of care after discharge.

The benefits of nurse-led face-to-face intervention could be enhanced by using instant messaging services or web support to improve medication adherence. Despite advancements in technology, these benefits remain unclear due to limited evidence, with only two studies showing efficacy available in our review [32,35]. Instant messaging services could be used to communicate with health professionals at an early stage. However, they should be followed by a telephone call or, ideally, an in-person meeting to address any issues. A Cochrane review found insufficient evidence regarding the effectiveness of mobile text messaging and internet-based interventions in promoting medication adherence [3].

Almost half of the studies observed a statistically significant improvement in medication adherence. However, it is still unclear whether variations in adherence scores in the scales used correspond to a real change in patient medication adherence [23,32,35,36,41]. In fact, although statistically significant, the differences in the adherence scores are minimal and do not meet the thresholds set by each scale for enhancing medication adherence. We observed, respectively, an increase of 5 points on a scale of 60 to 110 [35], 3 points on a scale of 5 to 25 [41], 1 point on a scale of 0 to 13 [36], and 1 point on a scale of 0 to 10 [32]. One study observed an improvement in adherence levels from low to medium, specifically focusing on patients who were considered poorly adherent [23]. The variation in adherence cut-offs and the lack of baseline measurements across studies do not allow a clear understanding of the effects of nurse-led intervention on medication adherence [21,27,30,40]. Additionally, in four studies, medication adherence increased in both the intervention and control groups, further complicating the attribution of the effectiveness to the interventions proposed [23,32,36,41].

Beyond the scale scores, it is important to consider adherence in conjunction with other clinical outcomes to maximize the clinical benefits of interventions. Studies that solely focus on adherence may not provide conclusive evidence of patient improvement. Indeed, changes in medication adherence alone may not necessarily reflect changes in other clinical outcomes, such as blood pressure, which could be influenced by various factors, including lifestyle changes. Similarly, studies that solely focus on clinical outcomes may not accurately assess the importance of adherence in achieving these outcomes [3]. According to Cochrane researchers, studies evaluating medication adherence should also assess the intervention’s impact on clinical outcomes to determine its true effectiveness [3].

In our review, while changes in adherence scores do not evidently support an improvement in medication adherence [23,32,35,36,41], the nurse-led intervention had a positive impact on blood pressure values [35,36]. However, these improvements only result in clinically relevant reductions in cardiovascular risk, defined as a reduction of 10 mmHg and 5 mmHg in systolic and diastolic blood pressure [56], respectively. In our systematic review, only one study achieved this goal, even if it had a small sample size [35].

### 4.1. Limitations

This review has several limitations. Firstly, we only included articles written in English. Secondly, the quality of reporting in some studies is poor, complicating the quality assessment. Thirdly, the sample size of studies included in the review varies greatly. Most of the included studies were relatively small, with the exception of two studies involving a larger number of participants [31,37]. Relatively small studies are more susceptible to bias. Fourthly, the description of the control groups was often inadequate. Many studies reported that the control group received standard care without providing detailed descriptions of the interventions, making it difficult to evaluate effectiveness. Additionally, in cases where standard care was already of good quality, improvements in medication adherence might not be observed. Many studies also lack sufficient details of the intervention, making it challenging to understand its effectiveness, application, and usefulness in other contexts. A further limitation of this review is the evaluation of the short-term effect of the nurse-led intervention, with only three studies having a 12-month follow-up period [23,35,36]. The heterogeneity in nurse-led interventions and outcome measurement meant that undertaking a meta-analysis was not possible; this could be considered a further limitation. Finally, the most widely used instrument for measuring adherence was the MMAS, in both older and updated versions. However, the validity of this scale has recently been questioned, discouraging its future use [57].

### 4.2. Risk-of-Bias Assessment and Interpretation

The evidence presented in this review should be interpreted with caution. The overall quality of the studies included is suboptimal, encouraging higher-quality research with a high methodological rigour that could identify the real effect of nurse-led interventions on clinically relevant outcomes. The efficacy of face-to-face interventions should be carefully appraised. In fact, excluding studies with a high risk of bias, only a minority of studies [23,26,35,41] with low risk of bias or some concern suggest that nurse-led face-to-face intervention could slightly improve medication adherence in chronic conditions, limited to cardiovascular and metabolic diseases. For these reasons, larger studies with long-term follow-up are needed. Considering the times of staff shortages, maximizing resources becomes even more important for health organizations. 

We would expect that nurse-led interventions need to be maintained for as long as the treatment is needed, integrating these in healthcare systems in a feasible and sustainable way. Investments in face-to-face interventions should be considered against the dedicated time required and the presence of highly competent nurses.

In regarding the quality of evidence carefully, RoB2’s most problematic domains were the first, the third, and the fourth. Specifically, thirteen studies did not provide full detail on the randomization process and the concealment of the sequence generation (domain 1). This may be due to limited word counts in journals and a lack of description of methods of randomization and allocation concealment, not necessarily meaning an inappropriate method. We carefully followed the RoB2 guidance and considered the presence of a central randomization as the minimum criterion for a judgement of adequate concealment of the allocation sequence. For future studies, a detailed description of the sequence generation is recommended to confirm the presence of a random component. The main reason for the negative impact on domain 3 assessment was the high percentage of studies with a consistent loss to follow-up of patients. According to the Rob2 guidance, we considered a proportion of less than 5% of the missing result data as “small”, which is considered small enough to exclude bias, and more than 20% as “large”. However, in all studies included in this systematic review, the author performed the recommended intention-to-treat analysis (ITT) to minimize this risk of bias. In addition, future studies could include sensitivity analyses to assess the potential impact of missing data. For domain 4, our assessment of the high risk of bias or some concern is due to the use of different questionnaires that have been not validated, and for this reason, they have a high probability of poor validity in detecting the outcome of interest. For future research, the use of validated questionnaires, possibly specific to the disease, must be encouraged to reduce this bias. 

### 4.3. Implications for Research and Nursing Practice

Research on medication adherence is a critical component of healthcare that requires improvement in various areas. Current methods of measuring medication adherence may not accurately capture patient behaviour and medication consumption patterns. Consequently, this represents a research area that could be enhanced. New methods for detecting medication adherence need to be implemented, focusing on assessing patients’ cognitive, emotional, and socioeconomic perspectives. Investigating the psychological mechanisms underlying poor adherence and its maintenance is important. Measurement of medication adherence is a challenge due to its complexity and the multiple factors that could affect it. A common method of measuring medication adherence is through patient self-report questionnaires. While these measures are widely applicable and easy to implement, the use of available validated questionnaires, possibly disease-specific, must be encouraged. Questionnaires used to assess adherence are often tailored to a single disease or provide general questions that may not capture the concerns in patients with multimorbidity. Assessment tools should also consider patients’ priorities, such as factors hindering their medication adherence. Future research on medication adherence should incorporate the patient’s perspective through qualitative studies to better understand common problems and difficulties experienced. Studies should also consider patients’ baseline levels of adherence and identify those who could benefit most from nursing interventions. Additionally, despite the increasing prevalence and complexity of these cases, there is a lack of research on interventions to improve medication adherence in patients with multiple chronic conditions. While interprofessional collaboration is valuable, the role of nurses should not be underestimated. Nurses have a unique level of closeness with and understanding of their patients that can foster trust in and adherence to treatment plans. In a future perspective, nurses can influence the allocation of resources to support the healthcare system and reach better patient health outcomes by increasing their policy power through their clinical knowledge and closeness to patients [58].

## 5. Conclusions

Our findings highlight the necessity of a targeted, durable and based-on-trust interaction between people with one or more chronic diseases and nurses to support and improve medication adherence. Building a trusting patient–provider relationship is an opportunity to discuss patient beliefs and concerns about the efficacy and safety of medications prescribed. In order to reach shared decisions, it is therefore necessary to explore not only the medication knowledge but also the psychosocial dimension of the patient. Nurses play a fundamental role in screening, assessing and supporting medication adherence and could coordinate the treatment initiation and future management.

The patient’s education is part of the nursing scope of practice that can be carried out through various techniques, including motivational interviewing, structured interviews, and assessment of care needs. Finally, technology provides valuable support for human actions but cannot be a substitute for them. Relationships and human interaction are still the main vehicles for educating and supporting people. A key element of improving health services is increasing nurses’ opportunities and capacities to participate in policymaking activities and subsequently manage chronic diseases.

## Figures and Tables

**Figure 1 healthcare-12-02337-f001:**
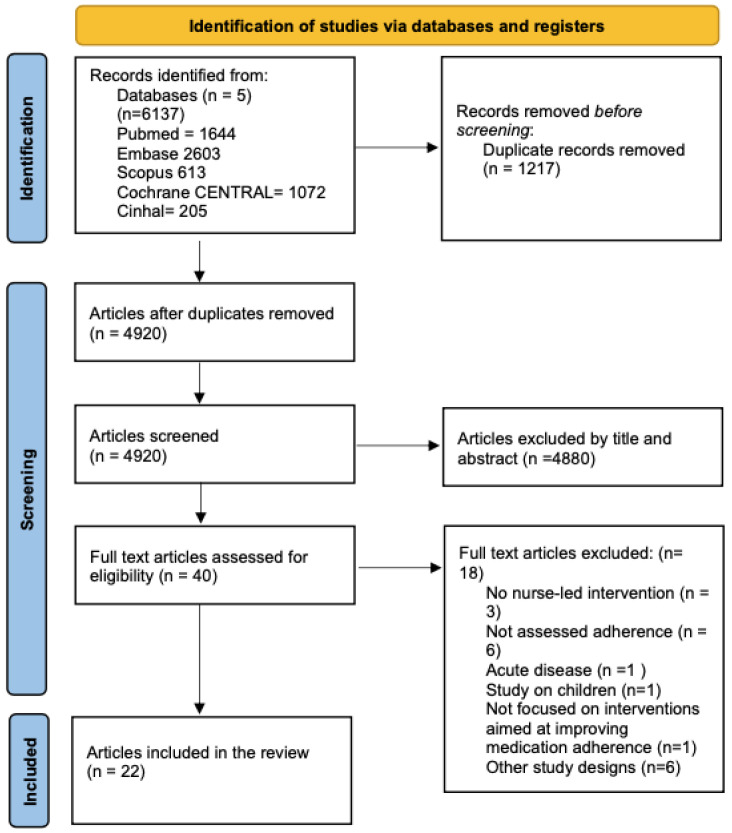
Flow diagram of the study selection.

**Figure 2 healthcare-12-02337-f002:**
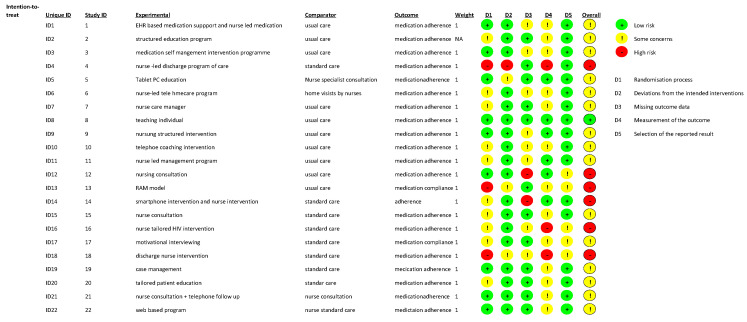
Bias assessment of the selected studies. Symbols and colours represent the risk-of-bias assessment for individual studies: “+” (green) indicates low risk of bias, “!” (yellow) indicates unclear risk of bias, and “-“ (red) indicates high risk of bias.

**Table 1 healthcare-12-02337-t001:** PUBMED search strategy.

PUBMED Search Strategy
(“Chronic Disease”[Mesh] OR “Comorbidity”[Mesh] OR “Polypharmacy”[Mesh] OR “chronic”[Title/Abstract] OR “chronical”[Title/Abstract] OR “chronically”[Title/Abstract] OR “chronicities”[Title/Abstract] OR “chronicity”[Title/Abstract] OR “chronicization”[Title/Abstract] OR “chronics”[Title/Abstract] OR “multimorbidity”[Title/Abstract] OR “comorbidity”[Title/Abstract] OR “polipharmacy”[Title/Abstract]) AND (“Nurse’s Role”[Mesh] OR “Nurse-Patient Relations”[Mesh] OR “nursing”[Subheading] OR “Nursing Process”[Mesh] OR nurs*[Title/Abstract] OR (“nurse-led”[Title/Abstract] AND “intervention”[Title/Abstract]) OR “nurse-led intervention”[Title/Abstract] OR “nurse-led care”[Title/Abstract] OR “Medication Review”[Mesh] OR “Continuity of Patient Care”[Mesh] OR “Tailored intervention”[Title/Abstract] OR “Health information technology”[Title/Abstract] OR “Telenursing”[Mesh] OR “Telemonitoring”[Title/Abstract] OR “Postdischarge follow-up”[Title/Abstract]) AND (“Medication Adherence”[Mesh] OR “Medication Therapy Management”[Mesh] OR “Self Care”[Mesh] OR “Self Care”[Title/Abstract] OR “Self-Management”[Mesh] OR “Self Management”[Title/Abstract] OR “Patient Compliance”[Mesh] OR “Health Behavior”[Mesh] OR “Patient Education as Topic”[Mesh] OR (“medication”[Title/Abstract] AND “adherence”[Title/Abstract]) OR “medication adherence”[Title/Abstract] OR (“patient”[Title] AND “compliance”[Title]) OR “patient compliance”[Title] OR “Patient Medication Knowledge”[Mesh] OR “Self Medication”[Mesh] OR “Drug Misuse”[Mesh] OR “Symptom Burden”[Title/Abstract] OR “Medication Safety”[Title/Abstract]) AND (“Clinical Trial”[Publication Type] OR “Controlled Clinical Trial”[Publication Type] OR “Randomized Controlled Trial”[Publication Type] OR “Clinical Trial”[Title] OR “Controlled Clinical Trial” [Title] OR “Non-Randomized Controlled Trial”[Title] OR “Randomized Controlled Trial”[Title] OR “RCT”[Title] OR “Non-Randomized Controlled Trial” [Title] OR “Quasi Experimental study”[Title] OR “Pre and Post Study”[Title] OR “Controlled Before-After Studies”[Title])

**Table 2 healthcare-12-02337-t002:** Summary of selected studies listed by date of publication.

Study ID	Author(s), Country, Year	Study Design	Setting	Participants	Intervention	Control	Medication Adherence	Adherence Measurement	Follow-Up	Main Results	Lost to Follow-Up	Risk of Bias
1	Holzemer, USA, 2006, [24]	RCT	Public HIV/AIDS clinic	HIV/AIDS	Structured educational intervention (118)	Usual care (122)	Primary outcome	Morisky Medication Adherence Scale (range 0–4, 0 very non-adherent, 4 very adherent); AIDS Clinical Trial Group-Revised Total Score (ACTG-Rev) (range 9–36, higher scores mean poorer adherence); Pill count (100% = perfect adherence); MEMS caps (100% = perfect adherence); Pharmacy refill records (100% = perfect adherence)	1 (T1)–3 (T2)–6 (T3) months	Morisky: IG % patients adherent: T0: 27.1%; T3: 30%; CG: T0: 25.4%; T3: 33.7% (*x*^2^0.61) (not significant); ACTG-Rev: IG % patients adherent: T0: 22.2%; T3: 23.2%; CG: T0: 28.3%; T3: 23.4% (*x*^2^ 1.18) (not significant); Pill Count: IG % patients adherent: T0: n/a; T3: 10.1%; CG: T0: n/a; T3: 12.6% (*x*^2^ 1.45) (not significant); MEMS caps: IG % patients adherent: T0: n/a; T3: 22.7%; CG: T0: n/a; T3: 20.9% (*x*^2^ 1.59) (not significant); Pharmacy refill records: IG % patients adherent: T0: 43.2%; T3: 33.7%; CG: T0: 25.4%; T3: 33.7% (*x*^2^ 0.40) (not significant)	27 IG 36 CG	High risk
2	Chiu, Hong Kong, 2010 [29]	RCT	Hospital	Hypertension	Structured educational intervention (31)	Usual care (32)	Primary outcome	Medication adherence (dose, frequency, timing of taking anti-hypertensive medication) (score 0–3)	8 weeks	No statistically significant differences in median (IQR) values (IG: pre-test; 3 (2–3); post-test: 3 (3–3); CG; pre-test: 3 (3–3); post-test: 3 (3–3); *p* < 0.235)	1 IG	Some concern
3	Wong, China, 2010 [39]	RCT	Two renal centres of a hospital	Chronic kidney disease	Telephone follow-up (49)	Usual care (49)	Primary outcome	Number of days of non-adherence and degree of non-adherence (score 0–4 = very severe)	7 (T1), 13 weeks (T2)	No statistically significant differences in mean (SD) medication days (IG: T0: 0.27 (0.9); T2: 0.12 (0.6); CG: T0: 0.43 (1.3); T2: 0.18 (1.0); *p* = 0.63) No statistically significant differences in mean (SD) medication degree (IG: T0: 0.29 (0.6); T2 0.08 (0.3); CG: T0; 0.27 (0.6); T2 0.12 (0.3); *p* = 0.66)	No drop out	Some concern
4	Gould, USA 2011 [22]	RCT	Academic medical centre	Acute cardiac event with PCI	Discharge nursing intervention (64)	Usual care (65)	Primary outcome	A modified Morisky Medication Taking Scale (MMAS-4) (5-point response options)	24 h after discharge	No statistically significant differences in mean rank [I = 61.55 vs. 68.39] (*p* = 0.266) No baseline data	25 (not specified in which group)	High risk
5	Lin, USA 2012 [34]	RCT	Primary care clinics	Diabetes, depression, coronary heart disease	Structured educational intervention (106)	Usual care (108)	Primary outcome	Automated pharmacy refill data in the 12 months before and after baseline *	6 (T1), 12 months (T2)	No statistically significant differences in mean (SD) values for each medication class Oral hypoglicemic: IG: T0: 0.83 (0.19); T2: 0.85 (0.17); CG: T0: 0.83 (0.20); T2: 0.83 (0.18). Antihypertensive: IG: T0: 0.85 (0.18); T2: 0.88 (0.14); CG: T0: 0.86 (0.18); T2: 0.88 (0.16). Lipid lowering: IG: T0: 0.82 (0.21); T6: 0.85 (0.17); CG: T0: 0.85 (0.18); T2: 0.88 (0.13). Antidepressant: IG: T0: 0.79 (0.23); T2: 0.85 (0.16); CG: T0; 0.80 (0.19); T2: 0.80 (0.19).	16 IG 17 CG	Some concern
6	Suhling, Germany, 2014 [26]	RCT	Hospital	Lung transplantation	Tablet computer-based patient education (32)	A nurse specialist (32)	Secondary outcome	Morisky Medication Taking Scale (MMAS-4) (range 0–4, higher scores means better adherence)	6 months (T1)	No statistically significant differences in mean (SD) values (IG: T0: 4 (0.25); T1 4 (0.18); CG: T0: 4 (0.34); T1: 4 (0.25) (*p* = 0.5)	2 IG 1 CG	Some concern
7	Granger, USA, 2015 [23]	RCT	Hospital	Chronic heart failure (“poorly adherent” MMAS score <6)	Structured educational intervention (44)	Usual care (42)	Primary outcome	Morisky’s Medication Adherence Scale (MMAS-8) (range 0–8, 8 = high adherence, 6–7.75 = medium <6 = low)	3 (T1), 6 (T2), 12 months (T3)	Mean (SD) adherence scores: IG: T0: 5.03 (1.41); T3: 7.04 (1.55); CG: T0: 4.8 (1.25); T3: 6.12 (1.33); *p* = 0.005	4 IG 7 CG	Some concern
8	Kekale, Finland, 2016 [25]	RCT	Eight secondary and tertiary care hospitals in Finland	Patients with chronic myeloid leukemia	Structured educational intervention (43)	Usual care (43)	Primary outcome	Morisky’s Medication Adherence Scale (MMAS-8) (range 0–8, 8 = high adherence, 6–7.75 = medium <6 = low)	9 months	Improvement of medication adherence in IG from low to medium or high rate in 17/35 patients ((49%) *p* < 0.0001) and in CG in 6/33 (18%) patients (*p* = 0.593)	8 IG 10 CG	High risk
9	Arruda, Brazil, 2017 [28]	RCT	Specialized clinic	Heart failure	Structured educational intervention (29)	Usual care (27)	Primary outcome	Brazilian Version of the Self-Care of Heart Failure Index Version 6.2. (range 0–26 points; higher scores indicate better adherence).	4 (T1) months	No statistically significant differences in mean (SD) values (IG: T0: 13.9 (3.6); T1: 14.8 (2.3); CG: T0: 14.2 (3.4); T1: 14.7 (3.5); *p* = 0.80)	18 IG 11 CG	Some concern
10	Persell, USA, 2018 [37]	RCT	Health centre	Hypertension	Structured educational intervention 2 study groups: EHR tool + plus nurse-led medication management support (278); EHR tool alone (262)	Usual care (254)	Secondary outcome	4-day assessment of pills taken and pills prescribed (full adherence vs. not)	3, 6, and 12 months	No statistically significant differences EHR tool + plus nurse-led medication management support vs. usual care: OR (95% IC): 0.9 (0.6–1.4); *p* = 0.59 EHR tool + plus nurse-led medication management support vs. EHR tool alone: OR (95% IC): 1.0 (0.6–1.5); *p* = 0.94	51 CG 40 EHR tools 35 EHR tools plus nurse-led education	Some concern
11	Cui, China, 2019 [30]	RCT	Hospital	Chronic Heart Failure NYHA II or III	Structured educational intervention (48)	Usual care (48)	Secondary outcome	The Chinese version of the Self-Efficacy and Health Questionnaire (range 0–20, higher score means better adherence)	12 months	Mean (SD) values IG = 15.3 (1.3) vs. CG = 12.9 (1.2) (*p* = 0.008) No baseline data	No drop out	Some concern
12	Mattei Da Silva, Brazil, 2019 [35]	RCT	Primary care clinic	Hypertension	Structured educational intervention (47)	Usual care (47)	Secondary outcome	The validated Questionnaire on Adherence to Treatment of Systemic Hypertension (scores range 60–110, lower score means poor adherence)	6 (T1)–12 (T2) months	Mean (SD) values IG: T0: 93.7 (5.8); T2: 98.4 (5.8) CG: T0: 94.9 (8.0); T2: 93.8 (6.9); *p* < 0.001	2 IG 2 CG	Some concern
13	Dwinger, Germany, 2020 [31]	RCT	Insurants registry	Type 2 diabetes, hypertension, coronary artery disease, heart failure, chronic depression and schizophrenia	Telephone-based health coaching (TBHC) intervention (1767)	Usual care (1222)	Secondary outcome	The “Medication Adherence Report Scale” (MARS-D), German version (range 5–25, higher score means better adherence)	12 (T1), 24 (T2), 36 (T3) months	No statistically significant differences in mean (SD) IG (T0: 24.01 (0.12); T3: 24.08 (0.12)) or CG (T0: 23.88 (0.12); T3: 23.92 (0.12)), *p* = 0.71	835 IG 614 CG	Some concern
14	Tessier, France, 2020 [38]	RCT	Ambulatory care clinic	Schizophrenia	2 study groups: smartphone intervention (SI-12) and nurse intervention (NI-11) (weekly telephone contact with patients)	Usual care (TAU: treatment as usual 10)	Primary outcome	Medication Event Monitoring System MEMS medication taking compliance (TAC) correct dosing (COD) timing compliance (TIC) **	6 months (T1)	No statistically significant differences between groups TAC: Mean (SD) T1 [TAU = 89.63 (14.84) vs. SI = 91.28 (12.30) vs. NI 93.78 (21.18) (*p* = 0.622)] COD: Mean (SD) T1 [TAU = 76.74 (25.79) vs. SI = 80.69 (13.42) vs. NI 82.88 (20.93) (*p* = 0.750)] TIC: Mean (SD) T1 [TAU = 65.73 (34.33) vs. SI = 70.07 (21.93) vs. NI 70.89 (31.59) (*p* = 0.813)] No baseline data	7 patients (not specified in which group)	High risk
15	You, China 2020 [42]	RCT	Hospital	Chronic heart failure	Telephone follow-up (84)	Usual care (74)	Primary outcome	Medications refilled in the electronic healthcare system	12 (T1) weeks	% of use angiotensin-converting enzyme inhibitor/angiotensin receptor blocker (ACEi/ARB): IG: T0: 81.3%; T1: 73.8%. CG: T0: 80.6%; T1: 59.7%. % of use beta-blocker: IG: T0: 72.5%; T1: 62.5%. CG: T0: 72.2%; T1: 51.4%. % of use aldosterone receptor antagonist: IG: T0: 61.3%; T1: 60%; CG: T0: 63.9%; T1: 54.2%; *p* < 0.05	4 IG 2 CG	High risk
16	Calvo, Spain 2021 [21]	RCT	Tertiary care hospital	Myocardial infarction	Structured educational intervention (68)	Usual care (75)	Primary outcome	Medication Taking Scale (MMAS-4) (one answer wrong = non-adherent); Haynes–Sackett test (taking tablets >80% = adherent); Pill count (not withdraw one medication box = non-adherent)	12 months (T1)	% patients adherent Morisky: IG: T1: 43/54 (79.6%); CG: 33/65 (50.8%), *p* < 0.001; Haynes–Sackett test: IG: 46/54 (85.2%) CG: 53/65 (81.5%), *p* = 0.391; Pill count: IG: 42/54 (77.8%); CG: 32/65 (49.2%), *p* = 0.002 No separated baseline data	14 IG 10 CG	Some concern
17	Hsieh, Taiwan, 2021 [32]	RCT	Medical centre	Atrial fibrillation	Web-based integrated management programme (116)	Nurse telephone follow-up (116)	Primary outcome	Medication Adherence Rating Scale (MARS) (ranges 0–10, higher score means better adherence)	1 (T1), 3 (T2), 6 (T3) months	Mean values (SD) IG: T0: 7.17 (1.79); T3: 8.5 (no SD) CG: T0: 6.97 (1.80); T3: 7.69 (no SD). *p* = 0.001	1 IG	Some concern
18	Liang, Taiwan 2021 [33]	RCT	Hospital	Multimorbidity	Nurse telemonitoring (100)	Usual care (100)	Secondary outcome	Chinese version of the Medication Adherence Behavior Scale (C-MABS) (range 6–24, higher score means better adherence)	3 (T1), 6 (T2) months	No statistically significant differences in mean (SD) values: IG: T0: 23.04 (2.08); T2: 23.79 (1.25) CG: T0: 23.13 (2.28); T2: 23.64 (1.13), *p* = 0.413	8 IG 19 CG	Some concern
19	Parra, Colombia, 2021 [36]	RCT	21 primary care centres	Hypertension, type 2 diabetes	Structured educational intervention (98)	Usual care (102)	Primary outcome	Treatment Behavior: Illness or Injury Questionnaire (range 0–13, higher score means better adherence)	6 (T1),12 (T2) months	Mean (SD) values: IG: T0: 9.40 (0.20); T1: 10.73 (0.20); T2: 10.43 (0.21) CG: T0: 9.38 (0.19); T1: 9.86 (0.20); T2: 10.03 (0.20) T1: *p* = 0.003; T2: *p* = 0.199	7 IG 7 CG	Low risk
20	Wu, China 2021 [40]	RCT	Hospital	Heart failure	Structured educational intervention (47)	Usual care (46)	Primary outcome	Morisky’s Medication Adherence Scale (MMAS-8) (score 6–30, 30 = 0 complete adherence, 25–29 = basic, <25 non-adherence)	1 month	% of complete adherence: IG = 61.70%; CG = 41.30%; *p* = 0.049 % of basic adherence rate: IG = 31.91%; CG = 34.78%; *p* = 0.769 % of non-adherence: IG = 6.38%; CG = 23.91%; *p* = 0.038 No baseline data	No drop out	Some concern
21	Zhang, China, 2021 [27]	RCT	Hospital	Hypertension	Structured educational intervention (60)	Usual care (60)	Primary outcome	Morisky’s Medication Adherence Scale (MMAS-8) (range 0–8, 8 = high adherence, 6–7.75 = medium <6 = low)	1 (T1), 2 (T2), 3 (T3) months	Mean (SD) values T3: IG = 6.57 (1.47); CG = 4.90 (2.16); *p* < 0.01 No baseline data	No drop out	High risk
22	Yang, China, 2022 [41]	RCT	Three community health centres	Hypertension, coronary heart disease, stroke cerebrovascular disease	Structured educational intervention (67)	Usual care (69)	Primary outcome	MARS-5—Medication Adherence Report Scale (Chinese version) (range 1–5, higher score means better adherence)	6 weeks (T1) and 3 months (T2)	Mean (SD) values: IG: T0 group: 15.43 (2.80) T1: 18.57 (3.23); T2: 17.88 (2.41) CG: T0: 15.70 (2.84); T1: 17.09 (3.39); T2: 16.98 (2.70). T1: *p* = 0.034 T2: *p* = 0.090	9 IG 20 CG	Some concern

* Automated pharmacy refill data in the 12 months before and after baseline to assess medication adherence by calculating percentage of days in the year that a patient obtained medicines from prescription fills divided by the number of days the patient should have been on the medication derived from a continuous, multiple-gaps-in-therapy method. Adherence was defined as the average for each prescribed medication class used to treat each disease parameter, weighted by the number of days within each observation window for each medication (i.e., the time between the first and last prescription fill). ** taking compliance (TAC): percentage of number of prescribed doses taken ((number of openings/number of prescribed doses) × 100); correct dosing (COD): percentage of days with correct number of doses taken ((number of days with number of openings as prescribed/number of monitored days) × 100); timing compliance (TIC): percentage of doses taken within prescribed interval ((number of openings within ± of 3 h around the prescribed interval/number of prescribed doses) × 100).

**Table 3 healthcare-12-02337-t003:** Characteristics of the nurse-led face-to-face and remote interventions.

Characteristics of Nurse-Led Interventions			
Face-to-Face Interventions		Delivery Methods	Timing of Follow-Up
Holzemer, 2006 [24]	A tailored, nurse-delivered intervention was designed to improve adherence to HIV/AIDS medications. The intervention’s content was based on a multifactorial framework for adherence proposed by Ickovics and Meisler. It evaluated the following areas with a standardized assessment: knowledge of medication taking, reasons for missing medications, use of medication reminders, self-reported adherence, medication troubles, medication side effects, role performance, and client–provider relationship.	Face-to-face visits at 1, 3, and 6 months and 3 telephone follow-ups in the week after the initial visit. The total time dose of the intervention ranged from 6 to 204 min.	1, 3 and 6 months
Chiu, 2010 [29]	A nurse clinic consultation and a telephone follow-up were performed, guided by a structured format: nurse self-introduction, general addressing of the patient’s health condition, adherence to a healthy lifestyle, reinforcing health self-management behaviours, providing health advice, and reviewing the mutually set health goals.	Two face-to-face visits that lasted about 45 min 8 weeks apart, and two telephone calls every 2–3 weeks during the span of 8 weeks.	8 weeks
Lin, 2012 [34]	A nurse care manager was responsible for enhancing patient self-management, responsiveness, continuity of care, systematic follow-up, and working with the primary care physicians. Nurse care managers identified patient-centred self-care goals and developed individualized care plans with problem-solving strategies.	Face-to-face visits or by telephone, initially 2–3 times per month.	6 and 12 months
Granger, 2015 [23]	A three-component intervention framework, based on medication bundles, symptom triggers, and the symptom response plan, was designed to support medication adherence. Patients participated in an in-depth, semi-structured interview to ascertain the prescribed medication regimen’s beliefs, concerns, and perceived necessity.	Face-to-face visits before discharge, and at 3, 6 and 12 months.	3, 6, and 12 months
Kekale, 2016 [25]	The intervention was based on tailored patient education combining nurse-conducted face-to-face counselling and interactive information technologies. The education session consisted of watching a 5 min video via an iPad at the hospital and a 30 min face-to-face counselling session with a hematology nurse based on the booklet and website information.	Face-to-face visit of 30 min.	9 months
Arruda, 2017 [28]	A combination of one-on-one nursing consultation and group meetings where nurses educated about the disease, lifestyle modification, and prevention and evaluated adherence and self-care maintenance, management and confidence.	Two face-to-face visits and eight group meetings over 120 days.	4 months
Persell, 2018 [37]	A combination of an electronic health record (EHR) tool (Medication List Review Sheet and a Medication Information Sheet) plus a nurse-led medication therapy management intervention from a nurse who identified areas for monitoring and follow-up with a teach-back method.	Face-to-face visits or by telephone (at least 1 medication educational session).	3, 6, and 12 months
Cui, 2019 [30]	A structured educational intervention based on two hours of educational sessions (one after symptom stabilization and one at discharge, based on the self-management theory by Norris et al.) aimed to reinforce knowledge of the disease and include self-care management measures, lifestyle modification strategies and medication compliance.	Face-to-face visits (one hour each) and telephone or face-to-face follow-up.	12 months
Da Silva, 2019 [35]	The intervention included nursing consultations, telephone contact, home visits, and group and individual health education activities. During the nursing consultations and home visits, the nurse case manager provided health education, measured blood pressure, checked the patient’s weight, and reviewed goals and healthcare plans, modifying them as necessary. Group activities focused on developing healthy habits, physical activity, treatment adherence, blood pressure measurement, and chronic complications.	Face-to-face visits were conducted every 6 months and lasted approximately 30–45 min, and telephone follow-up was caried out every 2 months and lasted approximately 5 min. Groups’ health education was conducted two or four times during 1-year follow-up, depending on the category risk of the patients, and lasted approximately 60 min.	6 and 12 months
Calvo, 2021 [21]	The intervention comprised home visits and reminder-type home calls at 6 months. The nurse detected patient needs and treatment problems during the home visits with a structured interview. The patient’s health education was personalized to increase therapeutic adherence as much as possible.	Face-to-face visits or by telephone at three months of admission. The duration of consultations was approximately 40 min.	12 months
Parra, 2021 [36]	The intervention included six educational sessions based on behaviour modification and coping enhancement. Participants received educational material.	Face-to-face visits, periodicity was monthly (six in total) and lasting between 20 and 40 min each.	6 and 12 months
Wu, 2021 [40]	Three Targeted Motivational Interviews (TMIs) were performed on days 2, 7, and 15 after hospital admission. The nurse formulated a plan for improving adverse behaviours together with the patient and set achievable goals.	Three face-to-face visits.	1 month after the first discharge
Zhang, 2021 [27]	The Roy Adaptation Model (RAM) was used to implement nursing plans based on physiological function, interdependence, role function, and self-concept.	Face-to-face visits during hospitalization.	Follow-up once per month after discharge, for a total of three times
Yang, 2022 [41]	A 6-week intervention consisting of three face-to-face educational sessions and two follow-up phone calls. Nurses used motivational interviewing techniques to help participants change negative attitudes and improve their self-management capacity.	Face-to-face visits (lasted approximately 30–40 min) and telephone follow-up.	Immediately post-intervention (six weeks), and at 3 months
**Remote Interventions**		**Delivery Methods**	**Timing of Follow-Up**
Wong, 2010 [39]	A telephone follow-up was provided in a structured format based on the Omaha system framework. The intervention consisted of the nurse’s self-introduction and general address of the patient, asking about the patient’s overall health condition, monitoring changes and progress from the specific health concerns, providing health advice, reinforcing health self-management behaviours, assessing the need for referral, and setting mutual goals.	Telephone follow-up every week for 6 weeks.	After completion of the 6-week disease management programme, and at 13 weeks
Gould, 2011 [22]	A discharge intervention was provided, consisting of written discharge materials (medication review materials, a medication pocket card, suggested Internet sites, copies of the interview tools) and telephone follow-up by an expert cardiovascular nurse.	Telephone follow-up.	24 h after discharge
Suhling, 2014 [26]	An iPad was used for education, with access to health education content and audiovisual materials. A single-page summary sheet was provided to take home. Educational content highlighted the importance of regular medication and its side effects and provided practical tips on achieving stable drug levels. A trained nurse specialist provided patient instruction using the designated written material in the conventional group.	Tablet Computer-based patient education. Face-to-face visits.	6 months
Dwinger, 2020 [31]	The intervention was based on counselling strategies and motivational interviewing (MI) to increase willingness to change and confidence to implement changed behaviours in daily life, individual and collaborative goal setting, and shared decision-making.	Telephone follow-up, with a minimum call frequency of one telephone contact every six weeks with a maximum intervention duration of one year.	Follow-up at 12, 24 and 36 months
Tessier, 2020 [38]	2 study groups 1. A smartphone-based intervention that administered daily medication reminders for one month, asking whether or not the patient had taken their medications, and then provided automated supportive statements to encourage adherence on days of medication non-use. 2. A manualised nurse-based intervention that provided telephone contact with patients to discuss potential medication use barriers and encourage adherence.	Telephone follow-up with weekly contact for one month.	6 months
You, 2020 [42]	During the first 14 days after discharge, nurse specialists called patients by telephone to ask about their conditions (e.g., clinical symptoms and signs of HF and body weight change), evaluate medication adherence, and provide immediate feedback.	Telephone follow-up	12 weeks
Hsieh, 2021 [32]	A web-based integrated management programme was designed, which includes five domains: patient information collection, instructions on atrial fibrillation knowledge, instructions on anticoagulation medicine, self-monitoring of symptoms, and professional consultation. Nurses provided telephonic coaching in the control group.	Tablet computer-based patient education Telephone follow-up thrice, at 1, 3 and 6 months	1, 3 and 6 months
Liang, 2021 [33]	The intervention consists of continuous telemonitoring through wireless transmission devices and home visits. The nurses composed personalized alerts set for each patient, and there was an open 24 h call centre. Nurses provided patients and caregivers with health education, nutrition and medication consultation, medication reminders, appointment scheduling, or follow-up reminders. Tele-homecare nurses also conducted home visits.	Telemonitoring and three home visits (at discharge, after 3 and 6 months).	3 and 6 months

## Data Availability

Not applicable.

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
