# Peer review of "Nurse-Led Interventions for Improving Medication Adherence in Chronic Diseases: A Systematic Review"

_healthcare, 2024, doi:10.3390/healthcare12232337_

Round 1
Reviewer 1 Report
Comments and Suggestions for Authors
The manuscript could benefit from a more thorough discussion on how the identified biases impact the overall interpretation of the results and possibly suggest ways to minimize these biases in future research.
The review should emphasize the need for standardized, validated adherence measures across studies, as the wide variability complicates comparisons and the interpretation of findings.
The discussion could include practical implications for healthcare systems and policymakers, particularly how nurse-led interventions might be integrated into routine care to improve chronic disease management.
Comments on the Quality of English LanguageNeed thorough proof reading for possible grammatical errors
Author Response
Point-by-point response to reviewer comments and suggestions for authors
Following a review by a native speaker with a clinical and academic background, extensive changes have been made to the text (highlighted for reference).
Comments/suggestions by reviewer 1.
- The manuscript could benefit from a more thorough discussion on how the identified biases impact the overall interpretation of the results and possibly suggest ways to minimize these biases in future research.
Thank you for your suggestion. After reviewing the studies and examining the RoB2 guidance, we added a specific section called “4.2 Risk of bias assessment and interpretation” (Pages 22-23, Lines 553-584) with the following paragraphs.
4.2 Risk of bias assessment and interpretation
“RoB2 domains with high risk or some concern assessments are domain 1, domain 3 and domain 4. Thirteen studies did not provide fully detailed details on the randomization process and the concealment of the sequence generation (domain 1). This may be due to limited word count in journals and a lack of description of methods of randomization and allocation concealment, not necessarily meaning an inappropriate method. In case of the randomization process is not properly due, the comparison of groups could be lost. If some prognostic factors affect the intervention group to which the participants are allocated, the estimated effect of the intervention will be biased by “conflicting”. Also, the presence of baseline imbalances could influence the measurement of results and undermine the reproducibility and generalisability of the results. Howewer, to reduce this bias, we carefully followed the RoB2 guidance and considered as minimum criteria for a judgement of adequate concealment of the allocation sequence the presence of a central randomization. For future studies, a detailed description of the sequence generation is recommended to confirm the presence of a random component.
The main reason for the negative impact on domain 3 assessment was the high percentage of studies with a consistent lost to follow-up of patients. According to the Rob2 guidance, we considered a proportion of less than 5% of the missing results data as “small”, which is considered small enough to exclude bias and more than 20% as “large”. However, in all studies included in this systematic review, the author performed the recommended intention-to-treat analysis (ITT) to minimize this risk of bias. In addition, future studies could include sensitivities analyses to assess the potential impact of missing data.
For domain 4, our assessment of the high risk of bias or some concern is due to the use of different questionnaires that have been not validated and for this reason, with a high probability of poor validity in detecting the outcome of interest. For future research, the use of validated questionnaires, possibly specific to the disease, must be encouraged to reduce this bias. Overall, the quality of the studies included in our review is sub-optimal and this finding should encourage the production of higher-quality research with a high methodological rigour to identify the real effect of nurse-led interventions, especially face to face, which require dedicated time and advanced nurses. In times of staff shortages, maximizing resources becomes even more important for health organizations”.
- The review should emphasize the need for standardized, validated adherence measures across studies, as the wide variability complicates comparisons and the interpretation of findings.
Thank you for your suggestion. We have tried to emphasize this concept by adding a specific sentence in the discussion.
Page 23, Lines 592-597: “Measurement of medication adherence is a challenge due to its complexity and the multiple factors that could affect it. A common method of measuring medication adherence is through patient self-report questionnaires. While these measures are widely applicable and easy to implement, the use of available validated questionnaires, possibly disease-specific, must be encouraged”.
- The discussion could include practical implications for healthcare systems and policymakers, particularly how nurse-led interventions might be integrated into routine care to improve chronic disease management.
We agree with you comment and consideration. We have integrated the discussion suggesting how nurse-led intervention could be implemented in healthcare services.
Page 20-21, Lines 471-490: “Involving nurses in advanced roles is a recognized way to optimize health care resources and increase quality of care. It is believed that the inclusion of nurses in a pivotal role can ensure that the demand for healthcare services that meet the needs of patients is adequately met [54]. Organizations such as the World Health Organization (WHO) have made several recommendations on how to strengthen the role of nurses (WHO 2012), and policy makers believe that in order to meet the challenges of hospitals and primary care, a more structured workforce in health care is required [55]. Nurse-led interventions lead to better health outcomes for a wide range of patient conditions and the empowerment of nursing role is vital and essential in the management of chronic diseases. With regard to the effects on clinical practice, despite the large heterogeneity of inventions, this study provides a wide view of nurse-led educational interventions that nurses could carry out both in community and hospital care. Practical implications for health systems and policy makers could be the integration of nurses-led clinics for the management of chronic diseases. The burden of complex and chronic diseases is increasing and requires expert nurses and financial resources to manage chronic and multimorbidity patients properly. Nurse-led clinics could ensure the management of people with chronic diseases and help identify patients with unmet needs or complications early, giving appropriate care for patients and reducing pressure on the health system. Nurse-led clinics could represent a bridge between the hospital and the community and ensure continuity of care after discharge”.
Reviewer 2 Report
Comments and Suggestions for Authors
The article is a systematic review of nurse-led interventions to improve adherence in adults with chronic diseases. The authors analysed 22 studies with 5,975 participants to assess the effectiveness of face-to-face, remote or mixed interventions. Face-to-face interventions showed statistically significant improvements in some studies, while remote interventions showed limited effectiveness. Methodologically, the authors adhered to the PRISMA guidelines and registered the protocol in the PROSPERO database. The search was comprehensive and spanned five databases with no time restrictions. However, in my opinion, the exclusion criteria could be more clearly defined. Also, when presenting the results, the table titles appear above the tables and not below. I would also recommend that the authors spell all acronyms in the "Notes" section of Table 1 to facilitate interpretation. The study does have some limitations, which the authors have carefully identified. Although the study has some limitations, which the authors were careful to point out, it nevertheless emphasises the role of nursing interventions in promoting treatment adherence.
Comments on the Quality of English LanguageThe English could be improved to more clearly express the research.
Author Response
Point-by-point response to reviewer comments and suggestions for authors
Following a review by a native speaker with a clinical and academic background, extensive changes have been made to the text (highlighted for reference).
Comments/suggestions by reviewer 2
- The article is a systematic review of nurse-led interventions to improve adherence in adults with chronic diseases. The authors analysed 22 studies with 5,975 participants to assess the effectiveness of face-to-face, remote or mixed interventions. Face-to-face interventions showed statistically significant improvements in some studies, while remote interventions showed limited effectiveness. Methodologically, the authors adhered to the PRISMA guidelines and registered the protocol in the PROSPERO database. The search was comprehensive and spanned five databases with no time restrictions. However, in my opinion, the exclusion criteria could be more clearly defined. Also, when presenting the results, the table titles appear above the tables and not below. I would also recommend that the authors spell all acronyms in the "Notes" section of Table 1 to facilitate interpretation. The study does have some limitations, which the authors have carefully identified. Although the study has some limitations, which the authors were careful to point out, it nevertheless emphasises the role of nursing interventions in promoting treatment adherence.
Thank you for your encouraging comments. We detailed the description of the exclusion criteria.
Page 4, Lines 161-165: “Studies that focused on addiction diseases, where adherence problems are typically of a different nature and much more severe, as well as studies reporting patients’ lived experience during nurse-led interventions and follow-up were excluded. In addition, conference proceedings, theses, letters to the editor, and other grey literature were excluded”.
All the acronyms are expressed in Table 1.
Reviewer 3 Report
Comments and Suggestions for Authors
Thank you for the opportunity to review this systematic review. Here are my comments:
-
Reasons are provided only for the exclusion of 18 articles excluded after full-text analysis. For articles excluded based on the title and/or abstract, only the number of excluded studies is given. I would like to see the criteria used for excluding studies, including the specific exclusion criteria and any categories of reasons for exclusion.
-
The journal logo header appears over Table 1.
-
Figure 2 is blurry and unreadable.
-
It might be worth mentioning in the limitations that a meta-analysis could not be conducted, especially since it was initially planned.
Author Response
Point-by-point response to reviewer comments and suggestions for authors
Following a review by a native speaker with a clinical and academic background, extensive changes have been made to the text (highlighted for reference).
Comments/suggestions by reviewer 3
Thank you for the opportunity to review this systematic review. Here are my comments:
- Reasons are provided only for the exclusion of 18 articles excluded after full-text analysis. For articles excluded based on the title and/or abstract, only the number of excluded studies is given. I would like to see the criteria used for excluding studies, including the specific exclusion criteria and any categories of reasons for exclusion.
Thank you for your suggestions. The PRISMA 2020 (Page et al. 2021; doi: 10.1136/bmj.n71) statement was adopted as reporting guidance for identify, select, appraise, and synthesise studies included in this systematic review.
According to the PRISMA statement, we did not specify in PRISMA flow diagram the specific exclusion reasons of each record we screened. However, the main reasons of excluding the articles were due to a different design of study, a lack of adherence measures, and study on acute diseases.
- The journal logo header appears over Table 1.
Thank you for your suggestion. We moved Table 1 under the journal logo.
- Figure 2 is blurry and unreadable
Thank you for your suggestion. We are sorry for this inconvenience and we have re-uploaded it.
- It might be worth mentioning in the limitations that a meta-analysis could not be conducted, especially since it was initially planned
Thank you for your suggestion. We added this paragraph in the limits of the study.
Page 22, Lines 547-549: “The heterogeneity in nurse-led interventions and outcome measurement meant that undertaking a meta-analysis was not possible; this could be considered a further limitation”.
Reviewer 4 Report
Comments and Suggestions for Authors
I would like to thank the editor for the opportunity to be part of the review process for this interesting manuscript. In addition, I congratulate the authors on their review work, but also on the relevant choice of topic.
As a systematic review of the literature, I have not identified any methodological weaknesses, so my recommendations for improvement will focus mainly on aspects of form.
In section 2.2, I suggest that the authors present the search strategy used in one of the databases in a table, referring to the others for supplementary material.
Due to its density, I suggest that table 1 be moved to ‘supplementary materials’ as an appendix to the manuscript.
All supplementary material should be in separate files from the manuscript, not integrated into the main file.
The authors have used PRISMA, so they should present the respective support tables as supplementary material.
Author Response
Point-by-point response to reviewer comments and suggestions for authors
Following a review by a native speaker with a clinical and academic background, extensive changes have been made to the text (highlighted for reference).
Comments/suggestions by reviewer 4
- I would like to thank the editor for the opportunity to be part of the review process for this interesting manuscript. In addition, I congratulate the authors on their review work, but also on the relevant choice of topic. As a systematic review of the literature, I have not identified any methodological weaknesses, so my recommendations for improvement will focus mainly on aspects of form.
Thank you for your encouraging comments.
- In section 2.2, I suggest that the authors present the search strategy used in one of the databases in a table, referring to the others for supplementary material.
Thank you for your suggestion. We presented one search strategy in a dedicated table (page 4).
An example of the applied search strategies is presented in Table 1.
Table 1 PUBMED search strategy
|
PUBMED Search strategy |
|
("Chronic Disease"[Mesh] OR "Comorbidity"[Mesh] OR "Polypharmacy"[Mesh] OR "chronic"[Title/Abstract] OR "chronical"[Title/Abstract] OR "chronically"[Title/Abstract] OR "chronicities"[Title/Abstract] OR "chronicity"[Title/Abstract] OR "chronicization"[Title/Abstract] OR "chronics"[Title/Abstract] OR “multimorbidity”[Title/Abstract] OR “comorbidity”[Title/Abstract] OR “polipharmacy”[Title/Abstract]) AND ("Nurse's Role"[Mesh] OR "Nurse-Patient Relations"[Mesh] OR "nursing"[Subheading] OR "Nursing Process"[Mesh] OR nurs*[Title/Abstract] OR ("nurse-led"[Title/Abstract] AND "intervention"[Title/Abstract]) OR "nurse-led intervention"[Title/Abstract] OR "nurse-led care"[Title/Abstract] OR "Medication Review"[Mesh] OR "Continuity of Patient Care"[Mesh] OR "Tailored intervention”[Title/Abstract] OR "Health information technology”[Title/Abstract] OR “Telenursing”[Mesh] OR “Telemonitoring”[Title/Abstract] OR “Postdischarge follow-up”[Title/Abstract]) AND ("Medication Adherence"[Mesh] OR "Medication Therapy Management"[Mesh] OR "Self Care"[Mesh] OR “Self Care”[Title/Abstract] OR “Self-Management”[Mesh] OR “Self Management”[Title/Abstract] OR "Patient Compliance"[Mesh] OR "Health Behavior"[Mesh] OR "Patient Education as Topic"[Mesh] OR ("medication"[Title/Abstract] AND "adherence"[Title/Abstract]) OR "medication adherence"[Title/Abstract] OR ("patient"[Title]AND "compliance"[Title]) OR "patient compliance"[Title] OR "Patient Medication Knowledge"[Mesh] OR "Self Medication"[Mesh] OR "Drug Misuse"[Mesh] OR “Symptom Burden”[Title/Abstract] OR “Medication Safety”[Title/Abstract]) AND ("Clinical Trial"[Publication Type] OR "Controlled Clinical Trial"[Publication Type] OR "Randomized Controlled Trial"[Publication Type] OR "Clinical Trial"[Title] OR "Controlled Clinical Trial" [Title] OR "Non-Randomized Controlled Trial"[Title] OR "Randomized Controlled Trial"[Title] OR "RCT"[Title] OR "Non-Randomized Controlled Trial" [Title] OR "Quasi Experimental study"[Title] OR "Pre and Post Study"[Title] OR “Controlled Before-After Studies”[Title]) |
- Due to its density, I suggest that table 1 be moved to ‘supplementary materials’ as an appendix to the manuscript.
Thank you for raising this point. However, it is a widely recognised convention that the presentation of the studies included in a systematic review is provided in a dedicate table, depicting a clear overview of the different characteristics of each study and allowing for an immediate comparison of the studies. As you were the only reviewer asking for this amendment, we opted to keep this table in the second version of our manuscript. Nevertheless, if the Editors will consider moving this table in the supplementary material, we would agree.
- All supplementary material should be in separate files from the manuscript, not integrated into the main file.
Thank you for your suggestion. We separate all the supplementary material in another file.
- The authors have used PRISMA, so they should present the respective support tables as supplementary material.
We are sorry you did not find them. We have uploaded the support tables as supplementary material.